# During hippocampal inactivation, grid cells maintain synchrony, even when the grid pattern is lost

**Noam Almog[1], Gilad Tocker[1,2†], Tora Bonnevie[3‡], Edvard I Moser[3], May-Britt Moser[3], Dori Derdikman[1]\***

[1]Rappaport Faculty of Medicine and Research Institute, Technion – Israel Institute of Technology, Haifa, Israel; [2]Gonda Multidisciplinary Brain Research Center, Bar Ilan University, Ramat Gan, Israel; [3]Kavli Institute for Systems Neuroscience and Centre for Neural Computation, Norwegian University of Science and Technology, Trondheim, Norway

**Abstract** The grid cell network in the medial entorhinal cortex (MEC) has been subject to thorough testing and analysis, and many theories for their formation have been suggested. To test some of these theories, we re-analyzed data from Bonnevie et al., 2013, in which the hippocampus was inactivated and grid cells were recorded in the rat MEC. We investigated whether the firing associations of grid cells depend on hippocampal inputs. Specifically, we examined temporal and spatial correlations in the firing times of simultaneously recorded grid cells before and during hippocampal inactivation. Our analysis revealed evidence of network coherence in grid cells even in the absence of hippocampal input to the MEC, both in regular grid cells and in those that became head-direction cells after hippocampal inactivation. This favors models, which suggest that phase relations between grid cells in the MEC are dependent on intrinsic connectivity within the MEC.
DOI: https://doi.org/10.7554/eLife.47147.001

**\*For correspondence:**
derdik@technion.ac.il

**Present address:** [†]Department of Neurobiology, Northwestern University, Evanston, Illinois; [‡]Department of Neuromedicine and Movement Science, Norwegian University of Science and Technology, Trondheim, Norway

**Competing interests:** The authors declare that no competing interests exist.

## Introduction

The mechanism responsible for the emergence of grid cell periodicity is arguably one of the best-tested and best-established mechanisms, across neural circuits in the mammalian central nervous system, thanks to extensive testing and analysis (*Hafting et al., 2005*; *Moser et al., 2008*; *Derdikman and Knierim, 2014*; *Rowland et al., 2016*). Modeling work has suggested that either grid cells are generated intrinsically in the medial entorhinal cortex (MEC), for example by a continuous attractor network model (*Burak and Fiete, 2009*; *Couey et al., 2013*; *Fuhs, 2006*; *Giocomo et al., 2011*; *Moser et al., 2014*; *Zilli, 2012*) or alternatively have their properties form through an interaction with another region, such as the hippocampus (*Dordek et al., 2016*; *Kropff and Treves, 2008*; *Stachenfeld et al., 2017*). To dissociate between these possibilities, we re-analyzed data from *Bonnevie et al. (2013)*, who inactivated hippocampal input to the MEC, and found that under this condition, the grid pattern of individual grid cells deteriorated significantly or disappeared entirely. Here we investigated whether the firing associations of grid cells depend on hippocampal inputs. Specifically, we examined correlations in the firing times of simultaneously recorded grid cells before and during hippocampal inactivation, including grid cells that acquired head directional tuning during inactivation. Our analysis yielded evidence of network coherence in grid cells even in the absence of hippocampal input to the MEC.

# Results

We reanalyzed data from *Bonnevie et al. (2013)* in which grid cells were recorded before, during and after hippocampal inactivation (*Figure 1A–C*). A total of 301 well-separated cells were recorded in the MEC and parasubiculum across 40 sessions including pre-, during and post-hippocampal inactivation, with 2–18 cells recorded simultaneously per session. While the runs of pre- and post-inactivation were analyzed in their entirety, for runs during inactivation we used only the time period starting at 15 minutes or later, up to 45 minutes, during which the most data were available across all recordings. The analysis showed similar effects on grid behavior in longer trials as in the first 45 minutes; the average grid score was $-0.063 \pm 0.193$ for the first 15–45 minutes and $-0.051 \pm 0.180$ for the remaining minutes.

We searched for evidence of network activity between grid cells in the absence of hippocampal input. Consequently, we examined spike time correlations and spatial firing correlations between simultaneously recorded cells whose gridness was high under normal conditions, but deteriorated during hippocampal inactivation (*Figure 1D and E*). To select only cells with significant gridness before inactivation and minimal gridness during inactivation, we used a minimum grid score threshold of 0.5 pre-inactivation, and a maximum of 0.2 during inactivation. Doing the same analysis with

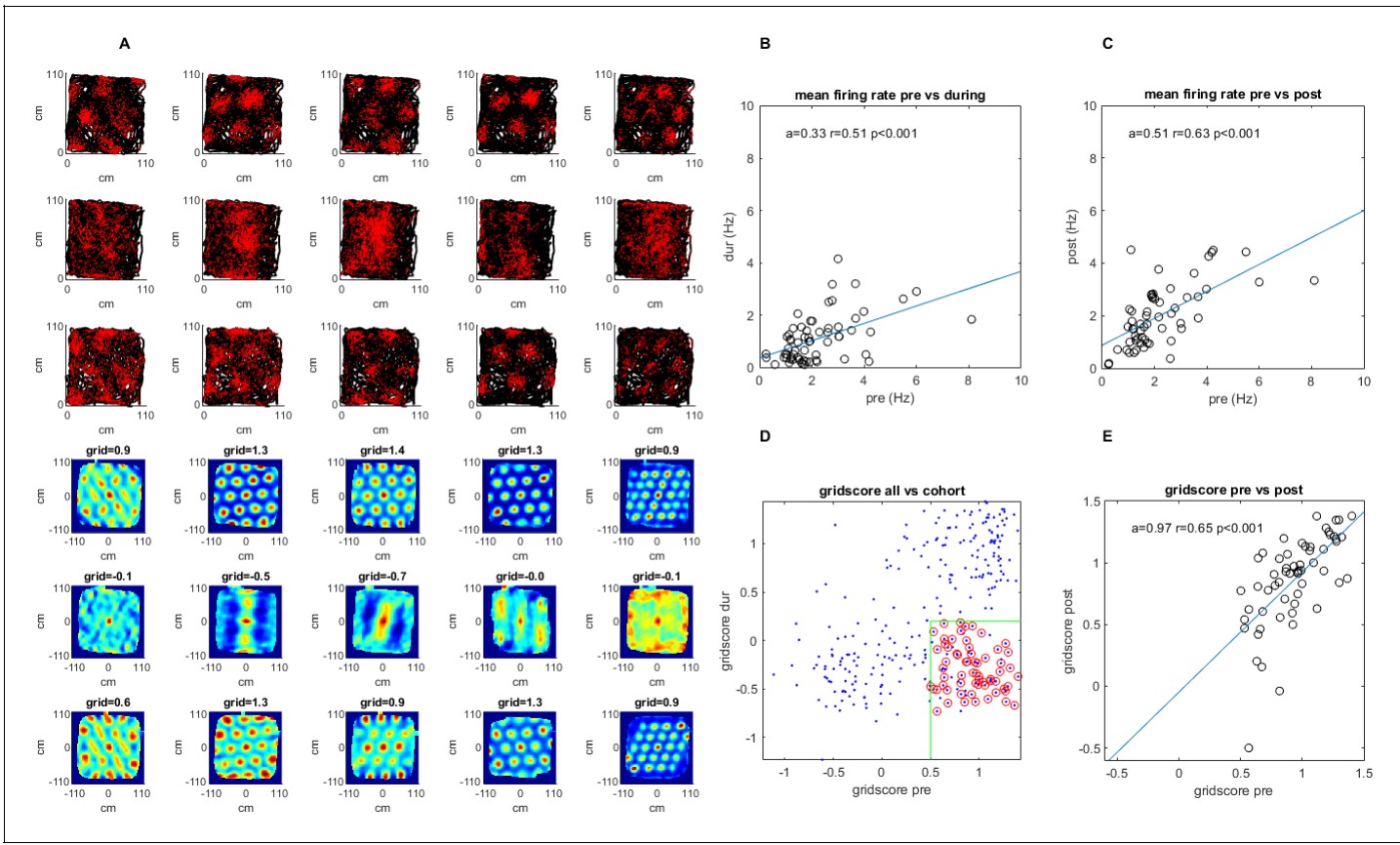

**Figure 1.** A survey of the grid cell population included in the study. Recordings were made pre-, during, and post-injection of muscimol to the hippocampus. (**A**) A sample group of 5 simultaneously recorded grid cells, one cell per column. The first three plots in each column show the location of a single cell firing (red) along the rat's trajectory (black) in a square arena pre-, during, and post-hippocampal inactivation, respectively. The last three plots in each column show the autocorrelation of the firing rate map and the grid score of that session pre-, during, and post inactivation. (**B**) The mean firing rate for the 63 grid cells included in the study pre- vs. during inactivation. (**C**) Same as (**B**) but for pre- vs. post-inactivation. (**D**) The grid score of all cells in the dataset vs. those included in the study. Red circles show the cells from the total population that were ultimately included in the study meeting the minimum and maximum grid score threshold pre- and during inactivation, respectively (green), as well as the additional criteria specified in the Materials and methods section (note that cells whose grid scores could not be calculated were set to 0). (**E**) Grid score pre- and post-inactivation of the cells included in the study.

DOI: https://doi.org/10.7554/eLife.47147.002

different thresholds, ranging from 0.2 to 0.9 pre-inactivation, and 0 to 0.4 during inactivation, did not change the central finding of the analysis, showing persistence of temporal correlations between cell pairs during hippocampal inactivation (*Figure 3—figure supplement 5*).

The mean grid score of the selected cells was 0.92 ± 0.24 pre-inactivation and −0.30 ± 0.24 during inactivation. Additionally, to ensure that the same cell was not recorded on different electrodes, we removed any cells from a single recording session whose individual spike times overlapped within a 1 millisecond window more than 5% of the time (this mostly removed cases with very large overlap that were suspected as originating from the same cell on different tetrodes). The mean spike overlap after exclusion was 0.57% ± 0.65 of the total spike strain. In total, 63 of 301 cells from 17 of 41 recording sessions met our criteria (*Figure 1D*), producing 107 pairs of simultaneously recorded cells, on which the results of this study are based. In our cohort, firing rates decreased by 50.0% during inactivation, and returned to 90.5% of original levels post-inactivation (pre 2.21 Hz ± 1.39, during 1.11 Hz ± 0.90, and post 2.00 Hz ±1.16). However, overall, neither the firing rate, nor the grid score seemed to correlate to temporal or spatial correlations both pre and during inactivation (*Figure 4—figure supplement 1*).

## Temporal correlations are maintained during loss of gridness

We found that several simultaneously recorded grid cell pairs consistently maintained temporal correlations even as their gridness score deteriorated (*Figure 2A and B*). Compared to random shuffling (n = 1000, α = 0.01), these correlations were statistically significant; 57%, 26%, and 53% of correlations passed the shuffling significance test pre-, during, and post-inactivation, respectively. Of the statistically significant correlations, 41%, 8%, and 27% were negative for the three respective recording phases, while 16%, 19%, and 22% were positive. Temporal correlations pre- vs. during inactivation were correlated at r = 0.58 (*Figure 3A*, p<0.001; although also in this case the distributions were different, Wilcoxon signed-rank p<0.001), and pre- vs. post-inactivation were correlated at r = 0.86 (*Figure 3A*, p<0.001). The results were very similar when analyzing the period of the recordings after 45 minutes (*Figure 3—figure supplement 1*). The strength of the correlations, both before and during inactivation, demonstrated a slight non-significant positive dependence on the grid score (*Figure 3—figure supplement 2*). For comparison, the correlation coefficient of temporal correlations pre- vs. during inactivation for each cell from our cohort against each cell not from the same recording session (1854 pairs in total) was r = −0.03 (p=0.21).

## Spatial correlations are maintained during loss of gridness

To examine whether short-range spatial correlations were maintained also during hippocampal inactivation, we compared the correlation coefficient of the 2D firing rate maps at the same position (x, y = 0,0). Overall, spatial correlations did not persist as consistently as temporal correlations during hippocampal inactivation; however, some degree of persistence was present between simultaneously recorded cell pairs. Spatial correlations were correlated to each other pre-inactivation vs. during inactivation at r = 0.34 (*Figure 3C*, p<0.001), while pre- vs. post-spatial correlations were correlated at r = 0.88 (*Figure 3C*, p<0.001). For a control comparison, the correlation coefficient of spatial correlations pre- vs. during inactivation for each cell from our cohort correlated against each other cell not from the same recording session, was lower (r = 0.10, p<0.001). The shuffling significance test (n = 1000, α = 0.01) found that 30%, 7%, and 22% of spatial correlations were significant pre-, during, and post-inactivation, respectively. Of the correlations, 18%, 1%, and 6% were negatively significant for all three recording phases, respectively, while 12%, 6%, and 13% were positive (*Figure 3B,D*). We note though that the amount of significant cell pairs was significantly lower during inactivation (χ2 = 12.2, p<0.001). The angular direction of the nearest peak in the spatial correlations was not maintained (*Figure 3—figure supplement 6*: while pre-inactivation vs. post-inactivation difference in angle was very non-uniform (Rayleigh non-uniformity test, p<0.001), the difference in angle between pre-inactivation and during inactivation did not diverge from uniformity (Rayleigh non-uniformity test, p=0.974), however when performing a time-windowed analysis of spatial correlations (see Materials and methods), at least in the 1 s range there were comparable correlations between spatial patterns before and during inactivation to those after inactivation (*Figure 3—figure supplement 4*). Although the statistical significance was lower overall for spatial correlations than for temporal correlations, the results for spatial correlations were consistent with those for temporal

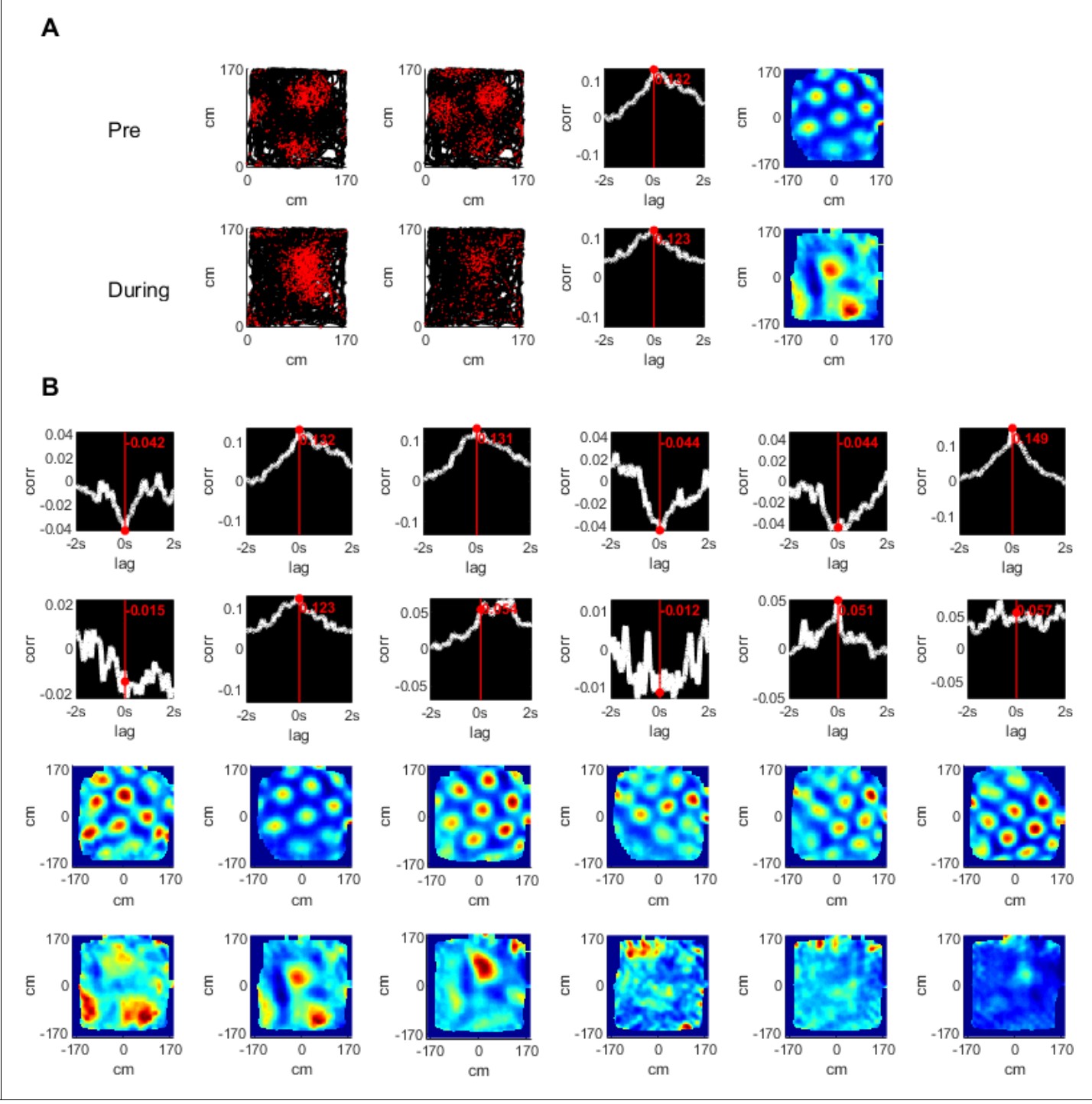

**Figure 2.** Temporal and spatial cross correlations of simultaneously recorded grid cells pre- and during hippocampal inactivation. (**A**) An example of a pair of simultaneously recorded grid cells (columns 1, 2); the location of the cell firing (red) plotted over the rat's trajectory (black) in a square arena. Columns 3 and 4 show the temporal and spatial cross correlation of the firing rate maps of the cells, respectively. Rows show the same analysis pre- and during inactivation. (**B**) The temporal and spatial cross correlations of cell pairs of an entire group of simultaneously recorded grid cells (one pair per column). Rows 1, 2 show temporal correlations pre- and during inactivation; rows 3, 4 show the same for spatial cross correlations.

DOI: https://doi.org/10.7554/eLife.47147.003

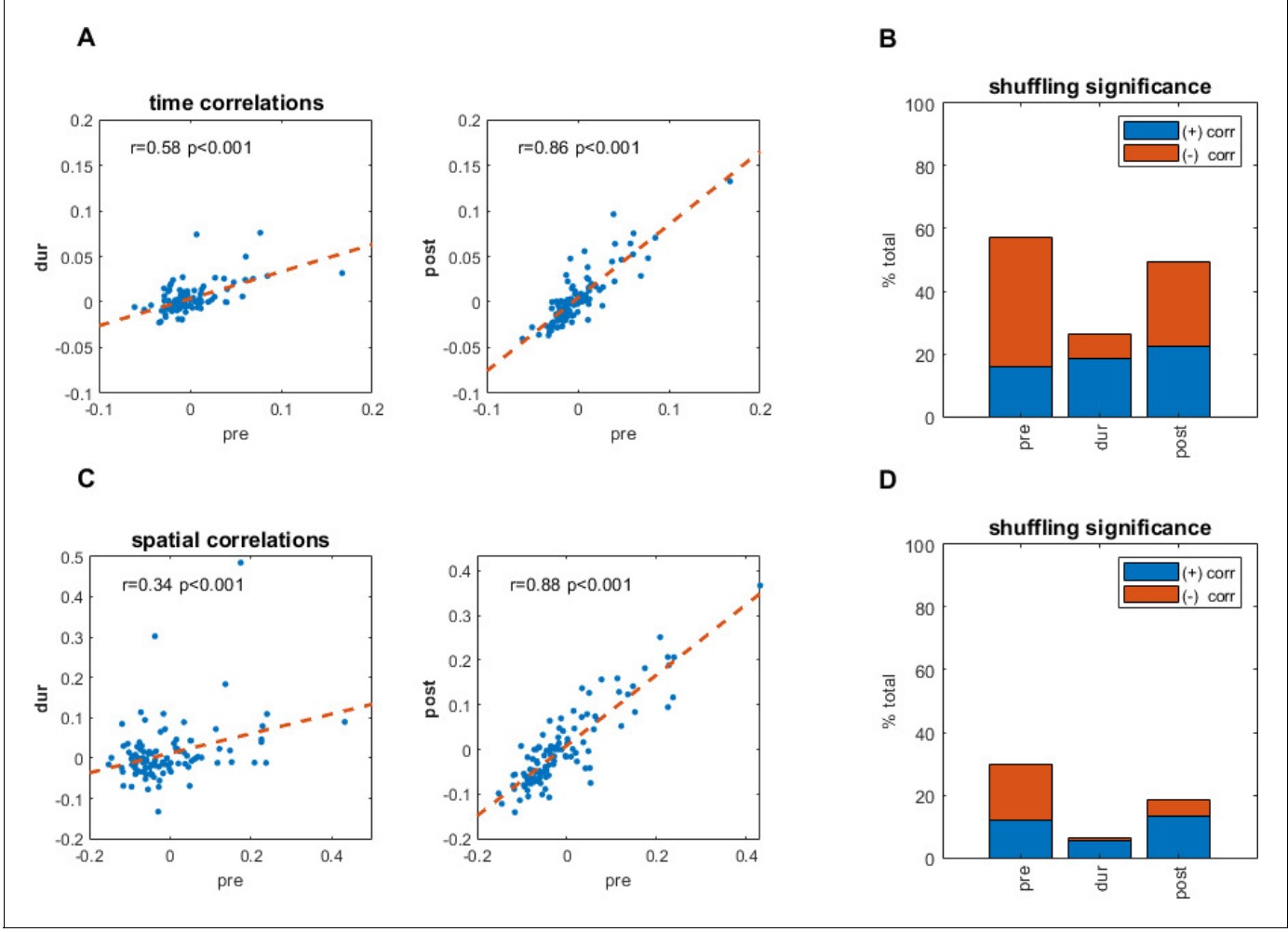

**Figure 3.** Temporal and spatial correlations pre- and during inactivation, for simultaneously recorded cell pairs. (A) Temporal cross correlations pre- and during inactivation, and pre- and post-inactivation, with correlation value (r), and corresponding p-value. (B) Proportions of significant temporal correlations, according to shuffling measures, pre-, during, and post-inactivation, including the sign (positive, negative) of the correlation value. (C) Same as (A) but for spatial correlations of the firing pattern in the arena at [0,0]. (D) Same as (B) but for spatial correlations of the firing pattern in the arena at [0,0].

DOI: https://doi.org/10.7554/eLife.47147.004

The following figure supplements are available for figure 3:

**Figure supplement 1.** Temporal correlations pre-inactivation plotted against the recording period during inactivation used in this analysis for cell pairs in cohort; all recordings after muscimol injections are from 15 to 45 minutes (top left) and all remaining recordings are from 45 minutes (top right) of the muscimol recording session.
DOI: https://doi.org/10.7554/eLife.47147.005

**Figure supplement 2.** Temporal and spatial correlations for cell pairs in cohort pre- and during inactivation plotted against their average grid score, including slope of the regression line (a), correlation coefficient (r) and p-value (p).
DOI: https://doi.org/10.7554/eLife.47147.006

**Figure supplement 3.** Temporal correlations plotted against spatial correlations for the cell pairs in our cohort, highlighted by significance, pre- (top) and during inactivation (bottom) with correlation coefficient (r) and p-value (p).
DOI: https://doi.org/10.7554/eLife.47147.007

**Figure supplement 4.** Drift rate plots for different time windows for all pairs in cohort.
DOI: https://doi.org/10.7554/eLife.47147.008

**Figure supplement 5.** Cell pair correlations between sessions and p-values as a function of the grid score thresholds used in cell pair selections.
DOI: https://doi.org/10.7554/eLife.47147.009

**Figure supplement 6.** Examining the angle of firing fields for cross-correlated cells.
DOI: https://doi.org/10.7554/eLife.47147.010

correlations. Additionally, plotting temporal and spatial correlations against each other demonstrated a clear linear relationship; the correlations of temporal to spatial correlations were r = 0.90 (p<0.001), r = 0.72 (p<0.001), and r = 0.90 (p<0.001) for pre-, during, and post-inactivation, respectively (*Figure 3—figure supplement 3*).

## Grid-turned head direction cells maintain temporal but not spatial correlations during inactivation

In the original *Bonnevie et al. (2013)* study, it was reported that a subset of the grid cells became head direction cells during hippocampal inactivation. Cells within our cohort showed a bi-modal distribution of Rayleigh head-direction tuning scores during inactivation (*Figure 4C*). This enabled selecting the cells that became head-direction cells, defined as having a Rayleigh score smaller than 0.4 pre-, and greater than 0.4 during inactivation (15 of the 63 cells in our cohort, from 3 of the 17 recording sessions in two rats; 37 pairs out of the 107 pairs) and those which did not, defined as staying below 0.4 both pre- and during inactivation (39 of the 63 cells in our cohort, from 13 of the 17 recording sessions in six rats; 46 pairs out of the 107 pairs) (*Figure 4C,D and E*). *Figure 4A* depicts an example of these grid-turned head direction cells from the same recording session. Examining these two clusters separately, looking at temporal correlations pre- and during inactivation, non-head direction pairs were more correlated at r = 0.71, than the head direction pairs at r = 0.52, though both were correlations were significant (p<0.001; *Figure 4D*). For spatial correlations, pre-inactivation compared to during inactivation, had a correlation coefficient of r = 0.46 (p<0.001) in the non-head directional cluster, and r = 0.22 in the head direction cluster, though this latter correlation was not significant with p=0.19 (*Figure 4D*). In conclusion, while spatial correlations were probably less persistent in grid-turned head direction cells, temporal correlations persisted similarly in both regular and grid-turned head direction cells, during hippocampal inactivation. There did seem to be a bias in the tuning angle for the grid turned head direction cells whose average was 37.0° ± 3.5 during inactivation, which was also seen in the rest of the cells in the population, whose Rayleigh score was also above the same threshold of 0.4 pre- and during inactivation. Their averages were 26.5° ± 6.1, and 38.7° ± 4.5 respectively (*Figure 4E and F*). This bias is consistent with the fact that all the grid-turned head direction cells were recorded in the same room (room 3), as were the majority of the rest of the cells (*Figure 4—figure supplement 2*).

## Discussion

This study reanalyzed data from *Bonnevie et al. (2013)*, in which hippocampal input to the MEC was inactivated. The aim was to examine possible evidence of local grid cell coordination. We found that despite the disappearance of the grid pattern of these cells during hippocampal inactivation, temporal correlations between grid cells remained, at least partially, as did local spatial correlations, although to a lesser, yet still statistically significant degree. A time-windowed version of the spatial correlations suggests a window of about 1 s in which spatial structure remained.

First, these findings assert that hippocampal input does not completely account for spatially and temporally correlated activity between grid cells. Second, the slow decay of the correlations at a timescale in the order of magnitude of 1 s indicates that this effect is not due solely to direct synaptic connectivity, but to recurrent network activity of dense networks in behavioral timescales. Last, some grid cells became head-direction cells during hippocampal inactivation, and their spatial correlations were less dominant; nonetheless, temporal correlations persisted. Taken together, this paper is adding another piece of evidence to a line of results that support continuous attractor dynamics (*Burak and Fiete, 2009*; *Couey et al., 2013*; *Fuhs, 2006*; *Giocomo et al., 2011*; *Moser et al., 2014*; *Zilli, 2012*; *Yoon et al., 2013*; *Heys et al., 2014*; *Gu et al., 2018*; *Trettel et al., 2019*; *Gardner et al., 2019*) while being less supportive of feedforward models creating grid cells through summation of information from the hippocampus (*Dordek et al., 2016*; *Kropff and Treves, 2008*; *Stachenfeld et al., 2017*). We cannot preclude, though, the possibility that grid cells are formed through a different feedforward process, not involving the hippocampus, or that they are generated through a recurrent loop involving information from both the hippocampus and the entorhinal cortex (*Donato et al., 2017*).

A network model for grid cell firing pattern at least strongly predicts, if not implicitly requires, a significant level of synchronicity in temporal firing between two connected grid cells (*Moser et al.,*

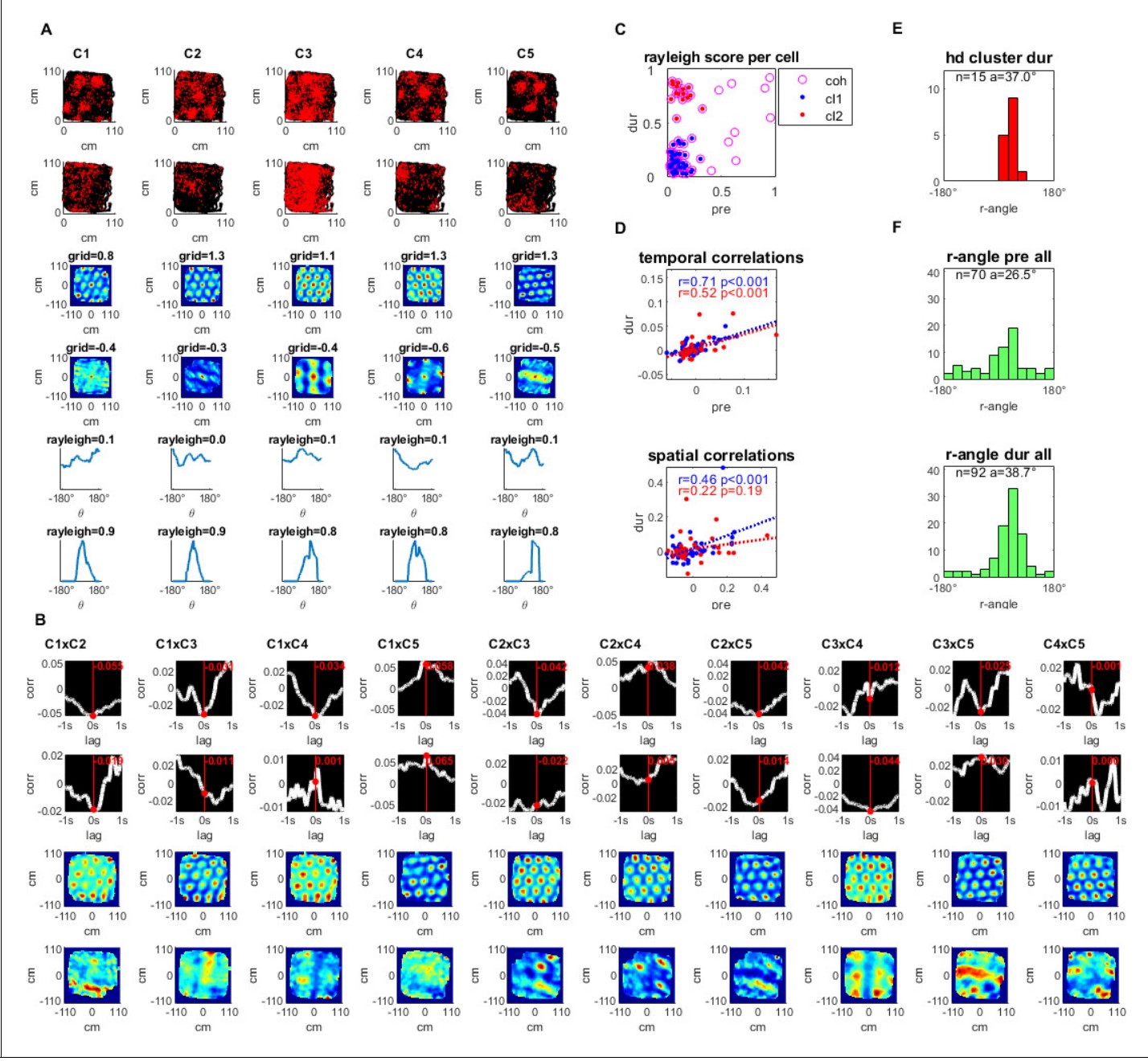

**Figure 4.** Simultaneously recorded grid cells that became head directional during hippocampal inactivation. (A) A sample group of 5 simultaneously recorded grid cells, one cell per column. The first two plots in each column show the location of a single cell firing (red) along the rat's trajectory (black) in a square arena pre- and during hippocampal inactivation. Next, two plots show the autocorrelation of the firing rate map and the associated grid score pre- and during inactivation. The last two plots in the column show the firing rate by head direction with an associated Rayleigh score, pre- and during inactivation. (B) Temporal and spatial cross correlations for each cell pair of the group, pre- and during inactivation by column. (C) Rayleigh scores pre- and during inactivation for all cells in the cohort (magenta circles) clustered by low head directionality (Rayleigh score <0.4 pre- and during inactivation, blue) and high head directionality (Rayleigh score <0.4 pre- and >0.4 during inactivation, red) (D) Temporal and spatial correlations (at 0,0) pre- and during inactivation grouped by head directionality clusters defined in (C), with the trendline slope (a) correlation coefficient (r), and corresponding p-value (p). (E) Histogram of Rayleigh angles for the cells in the HD cluster (red cluster in panels C and D) during inactivation (angles pre-inactivation are not shown since these cells had low Rayleigh scores for that period). (F) Rayleigh angles for all cells in population with Rayleigh score >0.4 pre-, and during inactivation.

DOI: https://doi.org/10.7554/eLife.47147.011

The following figure supplements are available for figure 4:

*Figure 4 continued on next page*

*Figure 4 continued*

**Figure supplement 1.** The mean firing rate of cohort cell pairs plotted against their mean grid score, temporal and spatial correlations, pre- and during inactivation (rows 1 and 2, respectively), grouped by cells with head direction selectivity during inactivation (Rayleigh score <0.4 pre- and >0.4 during inactivation, red), and without head direction selectivity during inactivation (Rayleigh score <0.4 pre- during inactivation, blue), and all pairs in cohort (black), with correlation coefficient (r) and p-value (p).
DOI: https://doi.org/10.7554/eLife.47147.012

**Figure supplement 2.** Rayleigh angles for all head direction cells in population by room number (rm#), for cells with Rayleigh score greater than 0.4, pre-, during and post- inactivation (rows 1, 2, 3, respectively), with number of cells (n) and average (a) +- standard deviation angle.
DOI: https://doi.org/10.7554/eLife.47147.013

*2014*). The attractor manifold model for grid pattern generation predicts that even when the network is deprived of spatial input, in this case from the hippocampus, the activity pattern that is maintained (though no longer anchored to physical space) would cause nearby cells in the manifold to fire with high correlation, and more distant cells not to fire (*Burak and Fiete, 2009*; *Dunn et al., 2014*; *Fuhs, 2006*; *Tocker et al., 2015*). This aligns with our observations of both correlated and anticorrelated activity during inactivation.

In addition to the evidence of network coordination, we found that the distinct subset of cells that became head direction-tuned during inactivation, maintained temporal but not spatial correlations during hippocampal inactivation. Interestingly, the head-direction selectivity of these cells was very biased (*Figure 4E*). The large overlap in Rayleigh angle between these cells enabled the temporal correlations to remain significant despite the transformation to head-direction cells. This subset of grid-turned head direction cells is likely defined by strong input from the retrosplenial cortex, or the pre- or para- subiculum, which also project into the MEC, and which dominate these cells' spatial tuning and firing time synchronization in the absence of input from the hippocampus (*Clark and Taube, 2012*). Because the grid-turned head-direction cells originated from recordings that did not contain grid cells that did not turn into head-direction cells we did not observe direct evidence of a temporal correlation between grid cells that became head-direction cells and those that did not. Notably, both groups were similar in their temporal correlation values before and during inactivation, regardless of their differences in spatial correlations. This suggests that despite receiving additional spatial input, the grid cells that became head-direction cells were part of the grid-cell network.

Several other experiments have investigated grid cell activity when spatial input was curtailed, specifically following removal of visual input. In their original paper that described grid cells, *Hafting et al. (2005)* observed rat grid cells in darkness, and found that the grid pattern did not deteriorate. More recently, two studies, by *Pérez-Escobar et al. (2016)* and *Chen et al. (2016)* that examined mouse grid cells in darkness reported that the grid pattern deteriorated without visual input. Furthermore, both studies found that significant temporal correlations were maintained during impaired spatial input to the entorhinal cortex, in accordance with our findings. Similarly, two new studies have demonstrated that grid cell temporal correlations remain during sleep (*Trettel et al., 2019*; *Gardner et al., 2019*).

The steady synchronicity in our study suggests an underlying network structure in the MEC that is responsible for grid cell formation. This corroborates the idea of an attractor manifold involved in grid cell formation.

## Materials and methods

The following sections describe the acquisition of the original data from *Bonnevie et al. (2013)*, and the analytical methods we applied to the data. All code was written in MATLAB (V. 2018b). The code was uploaded to GitHub at https://github.com/derdikman/Almog-et-al.-Matlab-code (*Derdikman, 2019*; copy archived at https://github.com/elifesciences-publications/Almog-et-al.-Matlab-code). The data files, in Matlab format, are available through Dryad at https://doi.org/10.5061/dryad.bk3j9kd6d.

## Input data

Briefly, the original experiment by *Bonnevie et al. (2013)* was performed with eight male, 3–5 month old Long-Evans rats, with water available ad libitum. The rats were kept on a 12 hour light, 12 hour dark schedule and tested in the dark phase. Rats were implanted with a microdrive connected to four tetrodes of twisted 17 μm platinum-iridium wire; one bundle was implanted in the MEC in all rats, anteroposterior 0.4–0.5 mm in front of the transverse sinus, mediolateral 4.5–4.6 mm from the midline, and dorsoventral 1.4–1.8 mm below the dura. Tetrodes were angled 10° in the sagittal plane. For hippocampal inactivation, cannulae were implanted at a 30° angle in the posterior direction towards the dorsal hippocampus; 0.24–0.30 μl of the GABA$_A$ receptor agonist muscimol (5-aminomethyl-3-hydroxyisoxazole) diluted in PBS was used to inactivate the hippocampus.

Rats were run in an open-field 100 cm, 150 cm, or 170 cm square arena, size depending on which of the three recording rooms were used (rooms 3, 9 and 10), polarized by a single white que card in an otherwise black environment for a 20 minute period, after which muscimol was infused. Subsequently, the firing rate of all principal cells recorded in the dorsal CA1 region decreased rapidly,~2.2 mm posterior and lateral to the infusion site (82 cells, all of which were place cells), with firing rates dropping to 1% of the baseline rate in 79% of the recorded cells within 20 minutes. Inactivation of the hippocampus had only minimal impact on the behavior of the rats. Rats then ran for an average of 160 minutes in the open field. After 6–24 hours, the rats were run for 20 minutes to check for cell recovery and grid stability.

## Quantifying gridness and head direction selectivity

For this analysis, we were interested in specifically examining grid cells whose spatial firing pattern was significantly degraded. To quantify this, we used the generally accepted measure of grid score, which essentially measures the extent the cell's firing pattern repeats itself at 60° intervals on a two-dimensional (2D) plane (how hexagonal the firing pattern is). The procedure undertaken to achieve this calculation is as follows (based on the procedure described in *Sargolini et al., 2006*; *Tocker et al., 2015*):

The arenas were divided into 50x50 equally sized square bins. First, a 2D map of neuron spiking was generated by creating a matrix where the index [i, j] represents the location in the arena, and the value represents the number of spikes in that location. The equivalent matrix for time spent at each location was also created. These two matrices were divided by each other element-wise, creating a matrix of firing rates at each location bin.

Next, a 2D normalized spatial autocorrelation was performed on the rate map matrix accounting for non-existent values in the rate map, as described in *Tocker et al. (2015)*. Firing fields were identified using a method that treated the smoothed (2D Gaussian smoothing with σ = 2) autocorrelation matrix as an image, and identified distinct regions bounded by a given pixel value of *x* in all eight directions whose external values are all less than *x*. Typical grid cell autocorrelations have at least six firing fields, at approximately 60° intervals (*Hafting et al., 2005*). Typical grid cell activity was manifested as equidistant firing fields at 60° intervals from each other.

The final step in calculating the grid score was to create a ring around the center of the smoothed autocorrelation (2D Gaussian smoothing with σ = 2), with an inner radius small enough to contain the innermost firing field, and the outer radius large enough to contain the outermost edge of the sixth closest field. Next, the ring was rotated 60° and correlated to the original (using a normalized correlation, accounting for empty matrix values as described above). This value was then subtracted by the value of the ring correlated at a 30° rotation. Since both correlations have values in the range of [−1,1], the range of grid scores is [−2, 2]. Cells whose autocorrelation did not create six distinct firing fields for calculating the annulus using the above method were set to a default grid score of 0.

A Rayleigh score from 0 to 1 was used to quantify head directionality of cells, similar to that described in *Tocker et al. (2015)*.

## Cell pair correlations

To quantify temporal correlations between cells, we calculated the Pearson correlation of their spike trains (lag = 0 ms). For spatial correlations, a 2D Pearson correlation of the rate maps (see Quantifying gridness) was performed and compared at [0,0]. Spatial cross correlations were done following the Pearson moment formula accounting for missing values in the rate map as described in

*Tocker et al. (2015)*. Smoothing was done on the spike trains prior to both temporal and spatial correlations using a moving average window of 25 ms. Varying the smoothing windows from 1 ms to 1000 ms had little or no impact on the correlation results.

## Shuffling to measure significance

To measure the statistical significance of the correlations, we employed a shuffling method in which spike train times were shifted cyclically n times by total_spike_train_time/n to create pseudo random spike trains (n = 1000 unless stated otherwise) and their correlations recalculated. Correlations were considered significant if they were in the 99[th] percentile when compared to the shuffled correlations.

## Time-windowed spatial correlation analysis

In order to measure spatial correlations at a smaller time-scale during inactivation, we analyzed spike to spike locations relative to the first cell in a cell pair within a given time window. For a given cell pair, each spike was treated as the origin in two dimensional space [0,0], and a 2D binned rate map (see above) consisting of all spikes in a given time window [±1s; ±2s; ±3s; ±5s or ±10s], was constructed. This collection of rate maps for a given cell pair was aggregated into a single rate map by adding up the values at a given bin location in the rate map. This procedure was done for each recording session [pre- during post-inactivation]. These cell pair maps were then spatially correlated against each other by session [pre- vs. during inactivation; pre- vs. post-inactivation], where [pre- vs. post-inactivation] represented the control in which we expected to see stable maps for cell pairs between pre -and post- and therefore high correlations.

## Acknowledgements

We thank Cindy Cohen for proofreading. We thank Chen Elbak and Irina Reiter for help with experiment administration. We thank members of the Derdikman lab for fruitful discussions.

## Additional information

### Funding

| Funder | Grant reference number | Author |
|---|---|---|
| Israel Science Foundation | 955/13,2344/16,2655/18 | Dori Derdikman |
| Rappaport Institute | | Dori Derdikman |
| Allen and Jewel Prince Center for Neurodegenerative Disorders of the Brain | | Dori Derdikman |
| Adelis Foundation | Technion-Weizmann collaboration grant | Dori Derdikman |

The funders had no role in study design, data collection and interpretation, or the decision to submit the work for publication.

### Author contributions

Noam Almog, Software, Investigation, Visualization, Methodology, Writing—original draft; Gilad Tocker, Software, Investigation, Methodology; Tora Bonnevie, Data curation, Writing—review and editing; Edvard I Moser, May-Britt Moser, Writing—review and editing; Dori Derdikman, Conceptualization, Supervision, Funding acquisition, Validation, Investigation, Methodology, Writing—review and editing

### Author ORCIDs

Dori Derdikman (iD) https://orcid.org/0000-0003-3677-6321

### Decision letter and Author response

Decision letter https://doi.org/10.7554/eLife.47147.018

Author response https://doi.org/10.7554/eLife.47147.019

# Additional files

## Supplementary files

• Transparent reporting form
DOI: https://doi.org/10.7554/eLife.47147.014

## Data availability

Original data for this study have been taken from Bonnevie et al. (2013). The link to the mat files is now available through Dryad at https://doi.org/10.5061/dryad.bk3j9kd6d. Analyses were performed in Matlab, and code was uploaded to GitHub: https://github.com/derdikman/Almog-et-al.-Matlab-code (copy archived at https://github.com/elifesciences-publications/Almog-et-al.-Matlab-code).

The following dataset was generated:

| Author(s) | Year | Dataset title | Dataset URL | Database and Identifier |
|---|---|---|---|---|
| Almog N, Tocker G, Bonnevie T, Moser E, Moser M-B, Derdikman D | 2019 | Data from: During hippocampal inactivation, grid cells maintain synchrony, even when the grid pattern is lost | https://doi.org/10.5061/dryad.bk3j9kd6d | Dryad Digital Repository, 10.5061/dryad.bk3j9kd6d |

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
