## [Decision Letter]

Thank you for submitting your article "During hippocampal inactivation, grid cells maintain their synchrony, even when the grid pattern is lost" for consideration by *eLife*. Your article has been reviewed by Laura Colgin as the Senior Editor, a Reviewing Editor, and three reviewers. The following individuals involved in review of your submission have agreed to reveal their identity: Ila R. Fiete (Reviewer #1); Kevin Allen (Reviewer #3).

The reviewers have discussed the reviews with one another and the Reviewing Editor has drafted this decision to help you prepare a revised submission. The individual reviews are included in their entirety toward the end of this decision letter for your information, but your rebuttal letter only needs to address the "Essential Revisions" listed below (i.e., you do not need to repeat your responses to individual reviewers' major comments).

Summary:

The authors show that patterns of correlation among grid cells observed prior to hippocampal inactivation persist during hippocampal inactivation. These findings are consistent with predictions of continuous attractor models, in which grid cell responses are predicted to be formed through intrinsic interactions within the grid cell network. The present paper demonstrates that the structure of correlated activity within the grid cell network, which has been reported in other papers, is not inherited from the hippocampus.

Essential revisions:

1) The paper should appropriately credit other previous works that have subsequently tested and found support for the continuous attractor hypothesis, as well as the key idea that the examination of cell relationships and their preservation across conditions where the inputs are modified is a key test of intrinsic pattern formation dynamics within the circuit.

2) Computing the correlations with different smoothing windows is unnecessary (Figure 2B). In this particular example, it is clear that correlation decays within more or less 1second, and that it is the only relevant timescale. This correlation can thus be captured by any choice of binning smaller than 0.5 seconds and the right filtering parameters (as in any sampling procedure). In addition, it cannot be concluded that this is consistent across time scales (as stated in the Discussion section). See the studies of HD/grid cell correlation during sleep for related analyses.

3) An additional analysis should be performed to address the question of which underlying spatial processes are preserved. One analysis that could be informative is to determine whether other pairwise spatial properties are maintained, e.g. the direction of the nearest peak in the spatial cross-correlograms.

4) The authors should investigate whether the grid scores of the cross-correlograms are higher when shorter bouts of exploration are analyzed. One possibility is that mutual grid patterns are preserved while individual grid cells lose their periodic structure because absolute phase drifts over time (see reviewer #2's comment below for specific analysis suggestions).

5) The study is presented as if it tested whether grid cells depend on local MEC circuits or external inputs (Abstract, Introduction and Discussion section). Because there are still many intact inputs to the MEC during hippocampal inactivation, this question can't be directly addressed with the current experiment. The authors should focus more on whether the firing associations of grid cells depend on hippocampal inputs.

6) In Figure 3A, the slope of the regression line appears to be less than 1. Was this significant or did one outlier cell pair only drive it? Is there a significant difference between the distribution of time correlations from "pre" and "during" (Wilcoxon signed-rank test)?

7) Figure 3B, there appear to be fewer significant cell pairs during inactivation (46% vs. 72%)? The authors should determine if this difference is significant. Since the recording time was longer during inactivation than for pre or post, one may have expected the opposite. Could this suggest that the grid cell correlation structure was not entirely stable during hippocampal inactivation?

8) A more thorough analysis of "grid cells turned HD" cells was requested. See reviewer #3's comment below for specific analysis suggestions. Also, the question of whether grid cells turned HD cells and pure grid cells were recorded in the same animals and whether grid cells turned HD cells were recorded in several animals.

9) It is not clear at which time the inactivation period starts. Is it immediately after injection or at the time of CA1 inactivation? Since 45 minutes of data is available, would it be possible to compare the firing associations during two non-overlapping periods (early vs. late inactivation)? This analysis would ensure that the correlations with the "pre" are maintained throughout the inactivation period.

*Reviewer #1:*

In this short, sweet, clear paper, Derdikman and colleagues examine the relationships between comodular grid cells before and during hippocampal (HPC) inactivation. They show that patterns of correlation among grid cells observed during spatial firing pre-inactivation persist during HPC inactivation, even in cells that lost most spatial tuning and turned into HD-like cells (at least temporal correlations are preserved for the HD-turned cells, while temporal and spatial correlations are preserved for the rest), consistent with the predictions of continuous attractor models in which cell responses are predicted to be largely formed through intrinsic interactions within the grid cell network rather than being inherited from another source like HPC input. This finding is at odds with the predictions of alternative models proposed by Treves, Derdikman, and others, in which the grid cell patterning is based on place cell input. It is definitely fit for publication in *eLife*, as it establishes a quantitative result, helps eliminate hypotheses, and contributes by further adding to accumulating support to the dominant hypothesis for the generation of grid cell responses. I also don't see any technical issues with the analyses, including the choice of cell selection criteria.

I do, however, have a suggestion that is closely related to my positive comments above, and is not unique to this paper, but is one that I would hope that this paper and subsequent ones by other authors following the example will follow. That is, it is unfortunately endemic to our field to start out papers by posing the question under study as one that has long been "mysterious" and "unanswered", and to pose the current work as the answer to this question. In this vein, the paper starts with "The means by which grid cells form regular, hexagonal spatial firing patterns has been an enigma since their discovery in the medial entorhinal cortex (MEC)." and "Since their discovery in the medial entorhinal cortex (MEC) (Hafting et al., 2005) the location and means by which grid cells form their eponymous hexagonal spatial firing patterns has been elusive." The current situation is not exactly this, however (though it once was!), as by now the grid cell mechanism is arguably one of the best-tested and best-established mechanisms, across neural circuits in the mammalian CNS, thanks to extensive testing and analysis. The current paper appropriately cites models that propose various grid cell mechanisms, but could do a much better job of appropriately crediting works that have subsequently tested and found support for the continuous attractor hypothesis, as well as the key idea that the examination of cell relationships and their preservation across conditions where the inputs are modified is a key test of intrinsic pattern formation dynamics within the circuit.

In sum, it will not diminish the paper's own results, and will rather greatly enhance the fidelity with which it represents our current understanding of grid cells, to state clearly in the Abstract and Introduction that this paper is adding another piece of evidence to a line of results that support continuous attractor dynamics, and citing these results. It's great that this is how the scientific edifice is built, and this work is a lovely contribution in that direction!

*Reviewer #2:*

Almog and collaborators have reanalyzed MEC recordings from Bonnevie et al., (2013) during which the hippocampus was transiently inactivated. While observing as in the original study a clear decrease in grid scores, the authors show that temporal and spatial coordination are maintained. They conclude that these observations support attractor (or at least local connectivity-based) models of grid cells. Overall, the idea and the results are interesting. The analyses could be improved a bit to strengthen the message.

Computing the correlations with different smoothing windows is unnecessary (Figure 2B). In this particular example, it is clear that correlation decays within more or less 1s, and that it is the only relevant timescale. This correlation can thus be captured by any choice of binning smaller than 0.5 seconds and the right filtering parameters (as in any sampling procedure). In addition, it cannot be concluded that this is consistent across time scales (as stated in the discussion). See the studies of HD/grid cell correlation during sleep for related analyses.

The fact that spatial correlations are maintained above chance level begs the question of which underlying spatial processes are preserved. One analysis that could be informative is to determine whether other pairwise spatial properties are maintained, e.g. the direction of the nearest peak in the spatial cross-correlograms.

Along the same lines, are the grid scores of the cross-correlograms higher when shorter bouts of exploration are analyzed? One possibility is that mutual grid patterns are preserved while individual grid cells lose their periodic structure because absolute phase drifts over time. One can also imagine to compute a time-limited spatial cross-correlogram: the idea would be to populate a 2D histogram of animal's positions at times of a neuron A spikes relative to the positions at times of neuron B spikes and within a window of +/- 1 second. This would correspond to a 2D histogram of all the pairs (xA-xB,yA-yB) for |tB-tA|<1s, with 'xi,yi' the position of the animal at times 'ti' of neuron 'i' spikes.

*Reviewer #3:*

This study by Almog and colleagues tests whether the correlation structure of grid cells is maintained after inactivation of the hippocampus with local muscimol injections. As previously reported (Bonnevie et al., 2013), grid cell periodicity and spatial selectivity are strongly reduced during hippocampal inactivation. In the current manuscript, the authors report that the firing associations between simultaneously recorded grid cells are maintained during inactivation.

This manuscript comes at the right time as two important studies addressing the stability of the grid cell correlation structure during sleep were just published. The results of the current manuscript provide convincing evidence that the firing associations between grid cells do not depend on hippocampal inputs.

One concern I have with the manuscript is that the study is presented as if it tested whether grid cells depends on local MEC circuits or external inputs (Abstract, Introduction and Discussion section). Because there are still many intact inputs to the MEC during hippocampal inactivation, this question can't be directly addressed with the current experiment. Perhaps the authors should focus more on whether the firing associations of grid cells depend on hippocampal inputs.

In Figure 3A, the slope of the regression line appears to be less than 1. Was this significant or did one outlier cell pair only drive it? Is there a significant difference between the distribution of time correlations from "pre" and "during" (Wilcoxon signed-rank test)?

Figure 3B, there appear to be fewer significant cell pairs during inactivation (46% vs. 72%)? Is this difference significant? Since the recording time was longer during inactivation than for pre or post, I would have expected the opposite. Could this suggest that the grid cell correlation structure was not entirely stable during hippocampal inactivation?

I find the results with "grid cells turned HD" cells interesting, but I am not sure how to interpret the findings. More analysis could be helpful here. For example, one could perform cross-correlations between the HD tuning curves during inactivation. Is map similarity during "pre" correlated with similarity in preferred HD during inactivation? In Figure 4A, the HD tuning curves during inactivation all look relatively similar (last row) despite the cells often having non-overlapping firing fields (top row). Theoretically, if grid cells are organized as a lattice with fixed connectivity, how can they become HD cells without modifying their firing associations with other grid cells? Would it be like trying to reorganize the grid cell lattice as a ring (typical HD attractor network)? Or do all grid cells turned HD cells inherit the same preferred direction during inactivation but fire at different times?

It is not clear at which time the inactivation period starts. Is it immediately after injection or at the time of CA1 inactivation? Since 45 minutes of data is available, would it be possible to compare the firing associations during two non-overlapping periods (early vs. late inactivation)? This analysis would ensure that the correlations with the "pre" are maintained throughout the inactivation period.

Were grid cells turned HD cells and pure grid cells recorded in the same animals? It would also be informative to know whether grid cells turned HD cells were recorded in several animals.

[Editors' note: further revisions were requested prior to acceptance, as described below.]

Thank you for resubmitting your work entitled "During hippocampal inactivation, grid cells maintain their synchrony, even when the grid pattern is lost" for further consideration at *eLife*. Your revised article has been favorably evaluated by Laura Colgin (Senior Editor), a Reviewing Editor, and two reviewers.

The manuscript has been improved but there are some remaining issues that should be addressed before acceptance, as outlined below:

*Reviewer #3:*

The authors have answered most of the concerns that I had raised during the initial review.

I still have one point that I think should be addressed. The authors now report that the distribution of temporal firing associations before inactivation is significantly different from that observed during inactivation (Figure 3A). Also, the number of significant temporal firing associations is lower during inactivation (Figure 3B). These two new findings should be given more considerations. Some grid cell pairs might maintain their firing associations (subsection “Temporal correlations are maintained during loss of gridness”), but as a population, there seem to be some modifications taking place in the firing associations during hippocampal inactivation. Perhaps the appropriate conclusion is that the firing associations (or synchrony) between grid cells are partially preserved during hippocampal inactivation. Maybe the title should be adjusted to reflect these findings.

---

## [Author Response]

Essential revisions:1) The paper should appropriately credit other previous works that have subsequently tested and found support for the continuous attractor hypothesis, as well as the key idea that the examination of cell relationships and their preservation across conditions where the inputs are modified is a key test of intrinsic pattern formation dynamics within the circuit.

We have now changed the first sentence in the Abstract: “The grid cell network in the MEC has been subject to thorough testing and analysis, and many theories for their formation have been suggested”.

Furthermore, we now use the reviewer’s words in the Introduction: “the grid cell mechanism is arguably one of the best-tested and best-established mechanisms, across neural circuits in the mammalian CNS, thanks to extensive testing and analysis.”

The relation of this work to attractor dynamics is now noted in the discussion: “this paper is adding another piece of evidence to a line of results that support continuous attractor dynamics”.

(Although we note in reservation that organized common synaptic input not from the hippocampus could also explain the results in the paper.)

Following the reviewers' comments, we have added the following citations to the paper: Yoon et al., (2013), Heys et al., (2014), Dunn et al., (2015), Gu et al., (2018), Trettel et al., (2019) and Gardner et al., (2019).

2) Computing the correlations with different smoothing windows is unnecessary (Figure 2B). In this particular example, it is clear that correlation decays within more or less 1second, and that it is the only relevant timescale. This correlation can thus be captured by any choice of binning smaller than 0.5 seconds and the right filtering parameters (as in any sampling procedure). In addition, it cannot be concluded that this is consistent across time scales (as stated in the Discussion section). See the studies of HD/grid cell correlation during sleep for related analyses.

Following this point, the analysis and discussion of different smoothing windows in the context of time scales has been taken out.

We now note in the Discussion section “the slow decay of the correlations at a timescale in the order of magnitude of 1 second” instead of the different smoothing windows, as was described before.

3) An additional analysis should be performed to address the question of which underlying spatial processes are preserved. One analysis that could be informative is to determine whether other pairwise spatial properties are maintained, e.g. the direction of the nearest peak in the spatial cross-correlograms.

We have now added the analysis of the direction of the nearest peak in the spatial cross-correlograms (Figure 3—figure supplement 6. As can be seen, comparing “PRE” vs. “During” to “Pre” vs. “Post”, the marked diagonal and long tail in the angle difference, seen in the latter case is seen less in the former (P<0.001 for the PRE-POST vs. non-significant for the PRE-DURING when comparing the angle difference to the uniform distribution). This analysis suggests that there isn't a strong relation between the direction of the peaks in the “Pre” condition and in the “During” condition in the cross correlograms.

4) The authors should investigate whether the grid scores of the cross-correlograms are higher when shorter bouts of exploration are analyzed. One possibility is that mutual grid patterns are preserved while individual grid cells lose their periodic structure because absolute phase drifts over time (see reviewer #2's comment below for specific analysis suggestions).

As suggested, we looked at the time-windowed spatial cross-correlations, trying out different time windows (1 second, 2 seconds, 3 seconds, 5 seconds and 10seconds). The procedure we used is now detailed in the description of Figure 3—figure supplement 4, and in the Materials and methods section.

In the 1 second time window case, we found examples of similar spatial cross-correlation patterns in the “PRE” vs. “During” condition. Comparing the correlations of these maps between the “Pre” and the “During” conditions vs. the “Pre” and the “Post” conditions suggests that in the 1-second window case (and not in longer windows) there was a significant correspondence in the correlations between those cases.

5) The study is presented as if it tested whether grid cells depend on local MEC circuits or external inputs (Abstract, Introduction and Discussion section). Because there are still many intact inputs to the MEC during hippocampal inactivation, this question can't be directly addressed with the current experiment. The authors should focus more on whether the firing associations of grid cells depend on hippocampal inputs.

This was now corrected in the Abstract and in the main text, as suggested.

6) In Figure 3A, the slope of the regression line appears to be less than 1. Was this significant or did one outlier cell pair only drive it?

Removing outlier changed slope from 0.41 to 0.50, still well below 1.

Is there a significant difference between the distribution of time correlations from "pre" and "during" (Wilcoxon signed-rank test)?

Indeed, the time correlations have a significant difference in the “pre” vs. the “during” phases (P<0.001, Wilcoxon signed-rank test). This is now mentioned in subsection “Temporal correlations are maintained during loss of gridness”.

7) Figure 3B, there appear to be fewer significant cell pairs during inactivation (46% vs. 72%)? The authors should determine if this difference is significant. Since the recording time was longer during inactivation than for pre or post, one may have expected the opposite. Could this suggest that the grid cell correlation structure was not entirely stable during hippocampal inactivation?

We performed a chi square test (1 DOF) and it showed that indeed the amount of cell pairs was significantly lower during inactivation (χ^2^=12.2, P<0.001). This is now noted in subsection “Spatial correlations are maintained during loss of gridness”.

8) A more thorough analysis of "grid cells turned HD" cells was requested. See reviewer #3's comment below for specific analysis suggestions.

Following reviewer’s #3 comment, we have now added a more thorough analysis of the head-direction cells in our data.

As pointed out by the reviewer, the Rayleigh direction of the grid-turned head direction cells was extremely biased. This panel is now in new Figure 4E.

We suggest that this huge overlap in the firing direction of the cells could potentially explain how pairs of cells could maintain their correlation even after they became head-direction cells.

We also looked at the direction of other head-direction cells in the population (classified as those with a Rayleigh score > 0.4 and more than 100 spikes). Surprisingly, these cells were also biased to some extent in all phases of the experiment (Figure 4F).

The unidirectional trend of the bias was recording-room dependent, as most of the cells were recorded from a single room (Room 3), Figure 4—figure supplement 2):

We note that also all 15 grid-turned HD cells were recorded in Room 3.

Thus, we believe the bias is related to an anisotropy in Room 3 (such as a cue card – we did not manage to check this, though).

These results are interesting and have been added to main Figure 4. The new results are now discussed in subsection “Temporal correlations but not spatial correlations persisted during inactivation for grid-turned head directional cells”, with an added sentence in the Discussion section.

Also, the question of whether grid cells turned HD cells and pure grid cells were recorded in the same animals and whether grid cells turned HD cells were recorded in several animals.

Grid cells turned HD cells were recorded in 2/8 animals from our cohort. In these two animals all grid cells turned into head-direction cells. This caveat is now highlighted in subsection “Temporal correlations but not spatial correlations persisted during inactivation for grid-turned head directional cells”.

9) It is not clear at which time the inactivation period starts. Is it immediately after injection or at the time of CA1 inactivation?

The inactivation period starts about 15 minutes after injection. Thus, all analysis was now updated to be aligned to the 15 minute start.

Since 45 minutes of data is available, would it be possible to compare the firing associations during two non-overlapping periods (early vs. late inactivation)? This analysis would ensure that the correlations with the "pre" are maintained throughout the inactivation period.

The figure comparing early vs. late inactivation has now been updated, demonstrating that the correlation phenomena exist also during the later phases of the inactivation: This was now updated in the paper as Figure 3—figure supplement 1.

[Editors' note: further revisions were requested prior to acceptance, as described below.]

Thank you for resubmitting your work entitled "During hippocampal inactivation, grid cells maintain their synchrony, even when the grid pattern is lost" for further consideration at eLife. Your revised article has been favorably evaluated by Laura Colgin (Senior Editor), a Reviewing Editor, and two reviewers.The manuscript has been improved but there are some remaining issues that should be addressed before acceptance, as outlined below:Reviewer #3:The authors have answered most of the concerns that I had raised during the initial review.I still have one point that I think should be addressed. The authors now report that the distribution of temporal firing associations before inactivation is significantly different from that observed during inactivation (Figure 3A). Also, the number of significant temporal firing associations is lower during inactivation (Figure 3B). These two new findings should be given more considerations. Some grid cell pairs might maintain their firing associations (subsection “Temporal correlations are maintained during loss of gridness”), but as a population, there seem to be some modifications taking place in the firing associations during hippocampal inactivation. Perhaps the appropriate conclusion is that the firing associations (or synchrony) between grid cells are partially preserved during hippocampal inactivation. Maybe the title should be adjusted to reflect these findings.

We agree that the message of the paper should be slightly changed.

This point has been updated in the conclusion of the Discussion section as follows:

“We found that despite the disappearance of the grid pattern of these cells during hippocampal inactivation, temporal correlations between grid cells remained, at least partially,…”

We also now write that: “these findings assert that hippocampal input does not completely account for spatially and temporally correlated activity between grid cells”

To account for this change, we have made a subtle change I the Title, by taking out the word “their”. The updated Title reads: “During hippocampal inactivation, grid cells maintain synchrony, even when the grid pattern is lost”.